# Towards Decolonial Choreographies of Co-Resistance

**Evadne Kelly * , Carla Rice and Mona Stonefish**

Re•Vision: The Centre for Art and Social Justice, College of Social and Applied Human Sciences, University of Guelph, Guelph, ON N1G 2W1, Canada; carlar@uoguelph.ca (C.R.)
* Correspondence: evadne@uoguelph.ca

**Abstract:** This article engages movement as a methodology for understanding the creative coalition work that we carried out for a project series called *Into the Light* (ITL) that used research from university archives to mount a museum exhibition and then develop an interactive public education site that counters histories and ongoing realities of colonial eugenics and their exclusionary ideas of what it means to be human in Canada's educational institutions. We address different movement practices, both those initiated by ableist-colonial forces to destroy difference and by our coalition of co-resistors to affirm difference. We apply a decolonizing and Anishinaabe philosophical lens alongside a feminist disability-informed neomaterialist and dance studies one to theorize examples of ITL's "choreographies of co-resistance". Anishinaabe knowledge practices refuse and thus interfere with colonial-eugenic practices of erasure while enacting an ethic of self-determination and mutual respect for difference. The ripple effect of this decolonizing and difference-affirming interference reverberates through our words and moves at varying tempos through our bodies—traveling through flesh, holding up at bones, and passing through watery, stretchy connective tissue pathways. These are our choreographies of co-resistance as actions of mattering and world-building.

**Keywords:** co-creation; decolonization; difference; disability justice; art; choreography; eugenics; Original Peoples; Anishinaabe; neomaterialism





This article engages movement as a methodology for understanding the creative coalition work that we carried out for a project series called *Into the Light* (ITL) that works in solidarity across our differences to counter histories and ongoing realities of colonial eugenics and its exclusionary ideas of what it means to be human in Canada's educational institutions. We situate our work in relation to the Anishinaabeg of the Three Fires Confederacy among the Ojibwe, Odawa, and Potawatomi upon whose ancestral lands our work takes place, and we honor, respect, and prioritize Original Peoples (First Nations) of this hemisphere and their worldviews as co-authors of ITL. The authors of this article are Mona Stonefish, Ogichidaakwe of the Anishinaabeg Three Fires Confederacy (Bkejwanong Territory), Doctor of Traditional Medicine, international activist for peace, women's and disability rights, and Grandmother Water Walker who was recently honoured by Chief Charles Sampson and Council of Bkejwanong First Nation for her thriving and determination as a residential school survivor, whose wisdom has guided ITL from the start, and who is receiving an Honorary Doctorate of Laws (16 June 2023) from the University of Guelph for her dedication to the pursuit of truth through centralizing the preservation of Anishinaabe language, culture, and tradition, advocacy for human rights, restorative justice, and decolonial education; Carla Rice, a crip-queer white settler critical embodiment scholar, who supervised Kelly's research and collaborated on ITL; and Evadne Kelly, a white settler dance-artist and scholar and project lead of ITL, whose curvy spine and large body regularly challenge normative constructions of the body upheld by western theatrical dance conventions. As justice-seeking artists, scholars, and activists who are invested in the power of the body and its movements to bring new possibilities for non-normative life into the world, we understand movement as encompassing the sensory, perceptual,

affective, emotive, communicative, cognitive, physical, psychic, spiritual, and all other actual and potential doings of bodies that ripple out into the world to have formative and transformative effects (Rice 2015, 2018; Rice et al. 2021b, 2022). The following draws on both Anishinaabe (or Nishnaabe to use Stonefish's and Michi Saagiig Nishnaabeg scholar-artist Leanne Betasamosake Simpson's spelling and pronunciation preference) and feminist disability-informed understandings to explore movement as a decolonial crip methodology for creative coalition work that resists collapsing difference and counters Canadian histories and ongoing realities of colonial eugenics. We call our movements together "choreographies of co-resistance" to foreground the agentic desires of bodies to live relationships marked by non-interference and respect for difference through mapping how these relations might materialize if we attend closely to process, place, time, and moving bodies (Foster 2011).

*Into the Light* aims to uncover histories of southern Ontario educational institutions producing and disseminating colonial "race betterment" (including eugenics) to trace those histories to present-day oppressions, and to counter them directly in our process and outcome. Southern Ontario is an important site for this work as it is often wrongfully assumed by many white, non-disabled middle-class settler Ontarians that race betterment belongs to another time and place, with Nazi Germany often thought of as the epitome of such violence. Instead, as central to the Canadian nation-state-in-formation's land occupation, southern Ontario has been an epicenter of white supremacist race betterment through strategic state interventions in families through forced and coerced sterilization and the forced placement of individuals from groups targeted for elimination in socially engineered residential environments including Indian Residential Schools, the so-called 60s Scoop, training schools, and reformatories. With knowledge of this history, the first author Kelly conducted research into university archives, organized a call for curators, and co-mounted an award-winning exhibition titled *Into the Light: Eugenics and Education in Southern Ontario* at the Guelph Civic Museum (Ontario Heritage Trust 2020). In addition to Kelly, exhibition co-curators included Stonefish, Peter Park (survivor of Oxford Regional Centre, co-founder of Respecting Rights, a project at ARCH Disability Law Centre, and founder of People First), Dolleen Tisawii'ashii Manning (Anishinaabe contemporary artist, philosopher, and Assistant Professor at Queen's University), Seika Boye (scholar, educator, dancer, and Assistant Professor Teaching Stream at the Centre for Drama, Theatre and Performance Studies, University of Toronto), and Sky Stonefish (Anishinaabe jingle dress dancer, photographer, and activist). Key collaborators and partners supporting and co-sponsoring the project included Rice (PI, Bodies in Translation), Sue Hutton (coordinator of Respecting Rights, a project led by people labeled with intellectual disability), and Dawn Owen (curator of Guelph Museums). *Into the Light*'s co-curatorial exhibition team made a conscious choice to prioritize stories of survivor-activists who have lived experiences of surviving and fighting eugenics. Stories they chose to share publicly are intimate and simultaneously protect their individual privacy. Since the knowledge of survivor-activists often becomes silenced and diminished, we asked visitors to engage with the stories as primary sources of expert knowledge of lived experience as well as the testimonies embedded in the stories that illuminate the histories and ongoing realities of systemic violence that comes from colonialism and eugenics. The exhibition incorporated artistic, sensory, and material expressions of memory in addition to archival artifacts as evidence of the role of education in accelerating and bolstering eugenics in Ontario.

Building on the exhibition's success, between May 2021 and August 2022, Kelly assembled another team to extend the reach of the knowledge generated through the survivor-activist testimony, her own ongoing archival research, and her learnings from audience engagement with the exhibition itself to create a new online learning platform entitled *Into the Light: Living Histories of Oppression and Education in Ontario*. To guide the development of this accessible, interactive, and multi-media platform full of artistic expressions of lived experience, we held 12 team meetings over eight months with Nishnaabeg, Métis, Black, racialized, white, disabled, labeled-as-disabled, verbal, non-verbal, and non-

disabled contributors. Music, visual art, and creative movement guided the development of our learning platform and our relational work together. At least half of our meetings featured guided movement activities led by Stonefish and Kelly as accessible, life-affirming, and decolonizing creation activities.

Our methodology of "choreographing co-resistance" works from the belief that socio-material realities engage and influence our embodied actions, which positions embodied action as a potent site for reproducing and/or intervening in the existing social order (Schneider 2015). Southern Ontario, as an early, sustained, and strategic site of colonial eugenics within which our embodied actions occur is imbued with sociopolitical and ecological significance for Anishinaabeg (Tuck and McKenzie 2015) and disabled peoples and thus is deeply relevant to our work. The term "co-resistance" centers the experiential wisdom of both Nishnaabeg and disabled survivor-activists who decentre the human-centric, unidirectional, and hierarchical ideas that agents of colonial eugenics have attempted to impose onto bodies and minds of difference. For us, choreographing co-resistance affirmed and altered our modes of embodiment without assimilating differences or imposing a normative standard during our ethically precarious creative processes of decolonial counter-eugenic political engagement.

Through the findings and makings that our methodology affords, we argue that choreographies of co-resistance using shared decision-making processes of self-determination and non-assimilation can create conditions of possibility through the collective generation of desires and practices for decolonization and accessibility. We believe that our coalitional experiences of collaborative research on ITL are emblematic of the complexly overlaid relationships that occur between bodies enacting settler and those enacting Anishinaabe worldviews, and between ableist-colonial and disability-affirmative decolonial orientations that unevenly center on and impact our moving bodies. By celebrating difference, the choreographies we co-create challenge ableist-colonial practices that aim to fold different embodiments into sameness and rank them against ableist-colonial aesthetic standards for bodily appearance and movement (Rice et al. 2021c). The choreographies that we surface in our learning platform push against the colonial-eugenic compulsion still evident in many university programs that involve the study of movement, including dance programs (Mitra 2021), kinesiology and physical education (Bailey et al. forthcoming a, forthcoming b; Bessey et al. forthcoming), public and preventative health care (Rice et al. forthcoming, 2015; Viscardis et al. 2019), and rehabilitation (Gibson et al. 2020).

In what follows, we offer three examples that demonstrate these choreographies of co-resistance that flow from different lived experiences of surviving and resisting colonial eugenics: the first focuses on the centrality of movement practices to the ITL exhibition and in the creation process of our learning platform. For our second example, we resurface selected passages of an 1844 debate that took place on Walpole Island (located in Bkejwanong Territory known today as southwestern Ontario) between Anishinaabe leaders who defended their sovereignty against Jesuit Missionary encroachment as local context to Anishinaabe understandings of difference. The debate shows how difference is welcomed from this Anishinaabe philosophical perspective because spirit is integral to everything in the universe, organic and non-organic. Spirit is not isolated to a transcendent god-figure that rules over and above humans and who imposes a hierarchy of life onto the material world. We bring this debate into dialogue with Stonefish's expressions of sovereignty during ITL to acknowledge and foreground the Anishinaabeg traditions of honoring the body, its movements, and its relationship to the universe (land, water, and sky or ecosystem). Revisiting this debate also allows us to attend to the force of Anishinaabeg ideas in countering Christian colonial language that seeks to diminish Anishinaabe knowledge and political orders, especially those regarding body–mind difference. Our third example builds on the two prior examples to show how Stonefish drew from her lived knowledge of Anishinaabe language, culture, and tradition and activated this knowledge in her approach to guiding us toward transformative understanding and action. Leading an ITL co-design session with a guided movement activity called "Touching the Universe with Love" Stonefish

encouraged participant-contributors to centralize our place-specific bodies, knowledge, traditions, and movements as unique sources of creativity within a communal process that rippled knowledge differently into being. The intention in reporting on our building of the learning platform in this way is not only to provide people with information about the locally specific operations/effects of colonial eugenics but also to provide some insight about the epistemologically and ethically charged processes of bringing to light the lived impacts of such life-destroying knowledge. This we worked to accomplish by creating a difference-affirming space that could foster and support a deeply felt-sensed engagement with ITL through responsiveness, connection, and accountability. To explore and theorize the three examples of ITL's "choreographies of co-resistance", we draw on research team discussions, personal correspondences between Stonefish, Rice, and Kelly, and we apply a decolonizing (Jimmy and Andreotti 2019) and Anishinaabe philosophical (Simpson 2011, 2017) lens alongside a feminist disability-informed neomaterialist (Barad 2006; Rice et al. 2021a, 2021b, 2022), and dance and performance studies one that disrupts a focus on conventional western theatrical notions of dance and performance (Foster 2011; Perazzo Domm 2019; Schneider 2015; Larasati 2013; Kelly 2019). Ultimately, our methodology prioritizes lived experience in our moving together as kinetics or doings that generate transformative and decolonizing learning within an ethic of non-assimilation that affirms differences and why they matter.

**Choreographies of Co-Resistance in Motion**

*Example 1: Into the Light Exhibit as a Choreography of Co-Resistance*

*Into the Light* focuses on exposing and resisting eugenics, a so-called "race betterment" theory and practice that sought and still seeks to control the future of humanity as part of a broader Euro-western intellectual and religious-imperial drive to control the direction of all earthly life (Kelly and Rice forthcoming). Over the 20th century, agents of race betterment, who included white settler researchers, politicians, doctors, social workers, educators, nurses, and other status figures prominent across social spheres, have embedded race betterment ideas and practices across institutions and institutional practices. These eugenic agents aimed to accelerate the cultivation of an ideal norm or "fit" "Canadian race", understood as white, Christian, non-disabled, hetero-patriarchal, and propertied, through two main strategies: providing those deemed as "fit" with incentives to reproduce, have families, and excel in life and work; and targeting those deemed "unfit", namely First Nations and racialized, poor, disabled, and labeled-as-disabled settler groups for dehumanization, diminishment, erasure, and elimination (Kelly et al. 2021a, 2021b; McLaren 1990; Stonefish et al. 2019). They created a hierarchy of mind over body to undermine embodied and experiential knowledge of groups that were largely excluded from the ranks of those deemed as knowers, leaders, and experts in institutions. For example, residential institutions (including Indian Residential Schools, training schools, poor houses, and asylums), organized and run by white religious and professional actors, took shape as immobilizing places of confinement and control that carried out ableist-racist aims of removal, incarceration, and ultimately, elimination of First Nations, racialized settlers, and settlers with body–mind differences, disabilities, and labeled-as-disabled (Ben-Moshe et al. 2014; Diverlus et al. 2020; McLaren 1990).

Despite ongoing resistance and the massive scientific and political discrediting of eugenics from the 1940s onwards due to the unprovability of its theories and the culpability of these in the atrocities of the Holocaust, eugenic praxis continues now, albeit in more stealthy, indirect, and insidious forms (Kelly and Rice forthcoming). Today, eugenic desires for race betterment continue to take the form of carceral logics, or practices of surveillance, regulation, discipline, and punishment embedded in and enacted by state systems (social welfare, healthcare, education, immigration), as abolitionists (Davis et al. 2022) and disability and transformative justice scholars point out (Ben-Moshe et al. 2014). Child welfare agencies continue to systematically take First Nations and Black children from their families and communities to be placed in white settler homes (Ontario Human Rights Commission 2018)

and medical practitioners continue to control human reproduction through, for example, forced and coerced sterilizations (Ataullahjan et al. 2021). Canadian colonial eugenics does not limit itself to control over human life: taking and controlling land and non-human life for settler benefit and profit comprises another eugenic impulse with disastrous ecological effects (Kelly and Rice forthcoming). These unjust and destructive betterment ideas and practices were woven into the building blocks of the Canadian nation-state, and, as such, continue to target bodies and lives (human and non-human) for dehumanization and elimination (Burch 2021).

The title of our work together, *Into the Light*, carries activist understandings of movement and emergent life that flow from difference-affirming and generating perspectives. From a Nishnaabeg perspective, Simpson understands dance, kinetics, or acts of doing, as "the crucial intellectual mode for generating knowledge", and as a methodology that enacts Nishnaabeg "embedded practices and processes" (Simpson 2017, p. 20) to animate knowledge that sustains diverse life. Referring to embedded practices of Nishnaabe-kwe that recuperate and revitalize knowledge specifically, Stonefish explains, "When we walk for the water, it reminds us of the sacredness of water and our responsibility to water. . . . We are the winds of change . . . traditions, language, and culture connect us to our land, water, and sky, *our* ecosystem, and that is what will truly sustain us" (Stonefish, pers. comm. with Kelly, 7 July 2021). Acknowledging the immobilizing and ecologically devastating effects of Canadian colonial eugenics, Stonefish uplifts the power of moving together to protect and care for earthly life. Thus, in their Nishnaabeg praxis, Simpson and Stonefish show how dancing and moving interactions and relations are world-making in the effects they generate, but in their respective projects, each reveals how these relations are also ethically complex networks or constellations of co-resistance that interfere in white settler colonial ideas/practices. Nishnaabeg knowledge systems generate knowledge through movement that encodes and animates that knowledge; and here Simpson and Stonefish turn to Nishnaabeg relations with land, water, and sky to activate the knowledge that is expressed and expanded through living relationships. These relations sustain balance and support the thriving of all life.

Neomaterialist theory also orients to how the physical world is constantly moving and, thus, aligns with Anishinaabe philosophy in understanding reality as ever-shifting and as continuously materializing differently in service of life's vitality (Rice 2014; Rice et al. 2021b, 2022). We draw on feminist settler physicist Karen Barad's understanding of reality as agentic based on her theoretical physics-informed understanding of diffraction or interference as a force that secures life's ongoing nature through its capacity to create difference. In physics, diffraction or interference comprises the idea that the energies that constitute the world do so through diffracting and interfering with each other (e.g., think about light, sound, and water waves bouncing off and changing the trajectories of each other) to re-produce existing or create new patterns and realities (Barad 2006; Rice et al. 2021b, 2022). Barad views diffraction not only as a metaphor but more so as the material practice of "making a difference in the world . . . taking responsibility for the fact that our practices matter; the world is materialized differently through different practices" (Barad 2006, p. 89). She understands knowledge, methodology, and ethics as entangled in how knowledge, and the research apparatus used to make it, changes reality, and, as she writes, "practices of knowing are specific material engagements that participate in (re)configuring the world". (Barad 2006, p. 91). This ethically entangled process of making knowledge and reality requires that we think with theories and design studies that have the potential to open space for difference to presence and to thrive (Rice et al. 2021b). Whilst materialist and Nishnaabeg thinkers orient to reality as processual, neomaterialists fail to acknowledge the centrality of relationality with the land to knowledge and worldmaking, or to ground their ethical frames in protecting and sustaining the vitalities and animacies of a place (Tuck and McKenzie 2015; Rice et al. 2021c). In response, Stonefish guided the ITL team to consider the significance of land- and place-based knowledge and experience. Knowledge made through creative movement relies upon the specifics of land and place to proliferate diverse

life, or difference, as opposed to producing sameness. During our online gatherings, disability activist-survivors Marie Slark and Antoinette Charlebois also shared movement and art practices that signified and generated hope, resistance, creativity, and self-determination in the face of physical confinement and carceral treatment in a residential institution for disabled and labeled-as-disabled children, youth, and adults called the Huronia Regional Centre. Through song, knitting, and needlework, they oriented to what research and creative practice might do for those of us who misfit with normative constructions of the human to generate culture, liberation, and change, and give greater access to life.

Both Anishinaabe and disability-informed neomaterialist approaches foreground movement as significant to knowing and worldmaking, as a methodology for interacting with and altering the world. We hold these differing theoretical traditions as complementary but not the same. Since the movements integral to each system have radically different roots and aims or desires—one in an Anishinaabe philosophical worldview concerned with the wellbeing of land and the universe, and the other in quantum physics interested in unlocking the nature of energy and matter—we work with them contiguously, in ways akin to Cree artist Elwood Jimmy's and settler scholar Vanessa Andreotti's braiding approach (Jimmy and Andreotti 2019). Mobilizing a braiding approach allows us to maintain the distinctions between, and integrity of, each knowledge system whilst materializing the differences in Nishnaabeg and crip experiences of colonial eugenics and in our sense-making of those experiences. We have used the term "interference" intentionally, for example, to not collapse differences in meaning, or appropriate or absorb different knowledge systems and worldviews into a singular framework as colonial systems have tried to do. We mobilize the term interference when we discuss the ableist-colonial drive to interfere with life to erase and eliminate difference. However, we also use the term in a different way. Rice et al. (2022) extend Barad's diffractive theory of the dynamism of reality to draw attention to the ways eugenic colonial logics have produced disability as a "lesser than" embodiment, and how biomedicine continues to interfere with the bodies and lives of those who misfit with its mythical norms in ways that still see difference as deficiency (Viscardis et al. 2019). The authors purposely mobilize the word interference rather than diffraction to jolt readers into awareness of the extreme violence done to misfitting body/minds within normative (health, education) systems and how those body/minds resist and re-orient to difference through disability cultures and activisms (Chandler et al. 2023; Rice et al. 2022). In addition to tracking such interferences put in motion by our motions, our methodology complicates neomaterialist notions of interference by foregrounding the Nishnaabeg ethic of noninterference, respect for self-determination, and difference as introduced by Stonefish and Simpson. We carefully hold the Nishnaabeg ethic of noninterference contiguous with the neomaterialist ethic that sees knowledge and ethics as always already entangled to think through our methodology—to think through how different knowledge systems might productively influence without interfering with or assimilating each other.

Importantly, respecting differences at an epistemic level does not mean we remain untouched and unchanged by each other (Rice et al. 2021a); rather, throughout the process of coming together to create ITL projects, we found ourselves opening to being touched and even changed by each other in non-directive and non-prescriptive ways through the movement practices we enacted to affirm Anishinaabe, crip, and other life and (even if fleetingly) (re)configure the world. Referring to our creative coalition or solidarity work together, we use the umbrella concept of "non-assimilation" to signal the ethic of respect for radical difference that emerges across our worldviews. We access and express that solidarity through the feelings and experiences that emerge from our creating together (Gaztambide-Fernández 2012; Rice et al. 2021a; Rice and Mündel 2018). We forefront the ethics that inform our methodological and actual dance together: the conscious, careful, and creative modes of knowledge generation and exchange that center mutual respect and non-assimilation in our decolonizing and accessible creation processes. We recognize that our encounters and configurations within this moving reality carry complex intersectional asymmetries, risk, and ethical precarity (Rice and Mündel 2019). ITL's movement activities, as "choreographies

of co-resistance", begin to articulate and map the complex interrelations between the physical and social worlds of ITL contributors and, in that articulation and mapping, ethically engage with land and place-specific socio-political and ecological realities that we, knowingly and unknowingly, reproduce (and disrupt) through our embodied actions.

Our individual understandings of what it means to move "Into the Light" are multifarious and reflect the ways in which difference has always been centered in our collaborative approach. For example, during the ITL museum exhibition, bodies moving through space created patterns of light and shadow. Light sources including focused overhead track lighting, projectors, backlighting, and lamps illuminated eugenic and counter-eugenic ideas. As visitors moved in and out of exhibition lights, their bodies interfered with light waves, and the light waves interfered with their bodies. Through movement, a person's body could block a eugenic idea from being projected into the space, but that also meant the eugenic idea appeared on their body. Bodies interfered with the light cast by projectors, and, in turn, bodies became sources of shadow and light, (potentially) prompting people to query how bodies and movements resist, interfere with, and become implicated in and impacted by the larger story being told (for detailed discussion, see Kelly et al. 2021a, 2021b). Entering the space, visitors moved with and through the rhythms and sound waves of Stonefish's voice delivering a traditional opening thanksgiving in Anishinaabemowin on a looped recording. The co-curatorial team configured her voice recording alongside a looped recording of a song by disability survivor-activist Peter Park. Together, their voices created a rhythm for visitors to move through the space with soundings celebrating difference. Through their vocal and rhythmic presence, Stonefish's and Park's bodies rippled out into the world to have formative and transformative effects on exhibition visitors.

In building our learning platform, we layered land- and place-based knowledge of water waves onto our work with light and sound waves as another source for understanding the movement interferences necessary for transformative decoloniality and disability justice. For example, the "Home" page of the ITL learning platform (go to: intothelight.ca) shows intersecting rivers at the site of our exhibition in Guelph, Ontario. Created by digital designers Jasmine Plumb, Shital Desai, and Ian Garrett in dialogue with the full team, the map of the Speed River joining with the Eramosa River creates three visual pathways in relation to eugenics taught at the Macdonald Institute and Ontario Agricultural College. The pathways center the lives of Stonefish (who survived the Mohawk Institute), Park (who survived Oxford Regional Centre), and Slark and Charlebois (who survived Huronia Regional Centre). The moving rivers show the intersections of colonial eugenics and resistance to it without folding lived experiences and journeys into one.

Through the energetic and sensory modes of learning offered by waves of light, sound, and water, Nishnaabeg living practices not only encode but also generate knowledge by "combining potential energy and kinetic energy as a creative force" (Simpson 2017, p. 183). Simpson's explanation of a "Stone's Throw" into water exemplifies the effects of doing/knowing/creating as acts of resistance with the potential to transform worlds:

[A]cts of resistance are like throwing a stone into water ... There is the original splash the act of resistance makes, and the stone (or the act) sinks to the bottom, resting in place and time. However, there are also more subtle waves of disruption that ripple or echo out from where the stone impacted the water .... Their path of influence covers a much larger area than the initial splash, radiating outward for a much longer period of time. (Simpson 2011, p. 145)

While Stonefish reminds us that "resistance comes at a very high price" (Stonefish, pers. comm. with Kelly, 29 March 2023), for Stonefish and Simpson, moving together to know-access-alter the world aims to affirm and embrace diverse life and to reject drives to determine and control the future as colonial eugenicists did. Such non-determining influence "matters in a dance towards decoloniality" (Mackinlay 2016, p. 220).

From a neomaterialist approach to knowledge making, visitors' movements in the Guelph gallery generated waves of energies that traveled, overlapped, interacted, and changed as they experienced the exhibition, itself produced within the walls of a museum

structure born as the Loretto Convent and residential academy. An inaccessible staircase adjacent to the source of Stonefish's voice served as a reminder that the place has been inflected, ordered, and oriented by the tensions that exist within a community impacted by and implicated in Christian colonial-eugenic histories and legacies. Counter to the rigid, exclusionary, and controlling normative ideas of bodies put in motion by the building's very structure, visitors from across and beyond Ontario moved in non-linear ways through the ITL exhibition to the rhythms set by Stonefish and Park in ways that made them more comfortable: some chose to sit directly on the floor, some leaned against walls, some sat on chairs provided by exhibition vignettes (thus implicating themselves in museum scenes), some relied on their mobility devices. The embodied and emplaced experiences and movements of visitors challenged the erasure of difference and put new patterns of thinking and feeling into motion—embodied actions that affirmed difference within local experiences. Bodies in motion were thus central to emergent knowledge. These examples show how we attended to the difference that a research practice makes in the world, using a methodological approach that "maps where the effects of difference appear" (Donna Haraway quoted in Mackinlay 2016, p. 213). These different perspectives regarding the movements of the physical world afford a non-hierarchical understanding of doing-knowing-creating through interaction, encounter, and interference. They invited a proliferation of stories and not a reduction of ideas; they activated visitors in ways we hoped would result from our work together; and they activated ripples that have traveled across and through bodies, which emerged as and with a new aliveness, as new entities unique to the encounters.

*Example 2: Moving with a Nishnaabeg Ethics of Difference*

The ethical and political complexities of undertaking coalitional research across embodied and embedded differences have reverberated throughout ITL. An ethical space of engagement is, as Cree scholar Willie Ermine writes, "formed when two societies with disparate worldviews, are poised to engage each other" (Ermine 2007, p. 193). For the ITL team, moving together is densely layered and entangled with ethical complexity and uneven power. For example, both scholars and survivor-activist team members embody lived experiences of settler colonial and eugenic interferences, resistance to such genocidal interference, as well as their own ethics of difference. Within this context, our work together across differences could be described as an "Im/possible Choreography" enmeshed within the paradoxical convergence of the possible with the impossible (Perazzo Domm 2019, p. 8). To unmake histories and legacies of oppressive and unethical contact, we needed to enter into relations or make contact through different tools and mechanisms on difference-centered terms (Mitra 2021). These have included a Nishnaabeg ethics of non-interference that upholds mutual respect, self-determination, and sovereignty (Simpson 2011, 2017) and a disability justice ethic that leads with disability to disrupt ableist practices and embrace difference-led decision-making and artistic approaches to research (Rice et al. 2018).

As our process unfolded, Stonefish brought forward her lived wisdom and the wisdom of her Anishinaabe ancestors on Walpole Island to deepen our collective understanding of Nishnaabeg ethics of non-interference and respect for difference and to consider what this understanding might mean for further decolonizing co-resistance. Prioritizing ancestral wisdom also centralizes Anishinaabe governance of the unceded territory, Anishinaabe presence and relations to the universe, and Anishinaabe efforts to challenge persistent settler notions of ownership over the land, water, and sky. Almost 200 years ago, the Anishinaabeg expressed resistance to western encroachment during a formal debate that took place in 1844 on Walpole Island between Ojibwe leaders and Jesuit Missionaries. The debate between Ojibwe leaders Chief Petrokeshig and 83-year-old warrior Oshawana representing thirty Elders present and Father Chazelle occurred within a context of increasing Christian colonial efforts to squeeze Original Peoples off their land and undermine their worldviews, and assimilate or eliminate them (Delâge and Tanner 1994; Krasowski 1999). In rejecting the establishment of a Christian mission on Walpole Island, Chief Petrokeshig spoke for

the Ojibwe leaders to refuse Chazelle's assertion of a single truth disconnected from the lived experience of difference, orating that

> ...the Great Spirit who made all things that exist made them with an infinite variety [ . . . ]. The trees are of many different species, and the plants are even more diverse. [ . . . ] *It is certainly the Great Spirit who put these innumerable differences in all that he created; consequently, his plan was not that we all have one and the same way of seeking the light.* I have a special way of seeking the light that he gave to me, you have yours that he also gave to you, and it is the same for all nations (quoted in Delâge and Tanner 1994, p. 311, emphasis ours).

Against Chief Petrokeshig's description of spiritual, epistemological, and material movements towards seeking difference, Chazelle argued that there was only one pathway to enlightenment, imposing a monotheist belief in a supreme all-knowing being that reflected the white man's image only and that denied the existence of other spiritual forces and pathways to truth:

> Once [the people of my island] accepted prayer they came to know the Great Spirit much better than you do, and since then they know that there is but one way to seek the light. [ . . . ] When anyone has Prayer, it is Jesus, the son of the Great Spirit made man himself, who teaches us wisdom by making himself known, by revealing himself. Yes, he reveals himself, not to the eyes of the body, but to the eyes of the spirit. [ . . . .] I need several feet of land, for my dwelling and the House of Prayer. The English to whom this island belongs, give them to me. (Quoted in Delâge and Tanner 1994, pp. 315–17)

By infusing spirit and truth in a singular supreme being that reflected only his image, Chazelle diminished embodied and lived experience as sites for knowledge generation and spiritual practice and refused the spirit, knowledge, and agency that the Anishinaabe understood as integral to everything in the universe. Further, his complete disregard for the Ojibwe leaders' assertion of their own intellectual and spiritual orders also justified the Jesuit's violent colonial interference with Anishinaabe sovereignty and governance. Yet Oshawana's final words reiterated the Ojibwe leaders' unwavering refusal of western encroachment and their demand for mutual respect for different worldviews and forms of worship.

> My brother, you have come to teach us that there is only one way, for all people, to know the Great Spirit and deserve his favors? [ . . . ] Except our way is different from that of White men. It must be that way, *because we see differences in all things*, according to the will of the Great Spirit. [ . . . ] Listen to what I have to say in conclusion. Stop building the cabin you have begun [ . . . ]. (Quoted in Delâge and Tanner 1994, pp. 319–20, emphasis ours).

To accelerate white settler supremacy, from the mid-19th to the mid-20th century, the Canadian state-in-formation increasingly used race betterment to form Indian Affairs Policy and the building blocks of the Indian Residential School system, striving to eliminate, assimilate, and suppress difference and dissent in its drive to become a white settler colonial society (Bednasek and Godlewska 2017). Despite ongoing resistance, the effects of Christian colonial interferences on the lives of Original Peoples have been and continue to be egregious. This was recently acknowledged by Pope Francis himself who, during his 2022 visit to Turtle Island to apologize for the church's role in Canada's IRS system, used the term genocide to describe its devastating effects on First Nations (Ka'nhehsí:io 2022). Condemning and eliminating First Nations language, culture, and traditions is, in the Pope's own words, a genocidal tool of colonization, or, in the words of Audra Simpson, a "sovereign death dance" (Simpson 2017, p. 115).

The Anishinaabe evidence their sustained resistance to colonialization in the documentation of the debate and in the way they have intergenerationally transferred the resistance and self-determination encoded in their language, culture, and knowledge (Krasowski 1999). For Simpson, the everyday embodiment comprises "a mechanism for ancient beginnings" that creates "flight paths out of colonialism and into magnificent unfolding" of locally specific resurgences of First Nations sovereignty (Simpson 2017, p. 193). As demonstrated by Stonefish, ancestral knowledge regarding self-determination, consent, and respect for difference on Walpole Island continues to be unwavering in the oratorical

teachings, worldviews, and traditions that have been carefully, strategically, and lovingly passed down and inherited. Engagement with these traditions continues within the wider political circumstances of ongoing settler-colonial eugenics. Stonefish explains: "Being human does not rely upon western modes of understanding. We need to counter such thinking and prioritize our language and culture in order to re-learn obligations of mutual care and to move forward together in a good way". By bringing forward the importance of the 1844 debate, Stonefish anchors her movements within a multigenerational choreography that remembers its traditions and refuses assimilation and genocide.[1] Beyond self-determination, bringing forth an understanding of embodied movement as inherent in learning and world-making within an ethic of non-assimilation and respect for difference, quoting Stonefish from our 25 August 2021 team meeting, "also connects us to our land, water, and sky (our ecosystem), which is what will truly sustain us into the future".

*Example 3: "Touching the Universe with Love"*

Aware of the violent effects of actions, or interferences, that impose singular and immobilizing truths, incorporating movement-based methods of interacting that prioritize difference requires a sensitivity to the relationships between movement and interference. We see decolonizing and difference-centered interferences across worldviews as generative—even and perhaps especially when they are generative refusals. As Simpson notes, centering generative refusal when building a movement that "pivots on (re)connecting with the land and with practices that sustain life and land" has "world-making effects" (Simpson 2017, p. 178). Such refusals include any one or a combination of movements as expressions of agency and self-determination, those that offer ways of resisting colonial eugenics, and those enacted as "deliberate and strategic resurgence", amongst others (Simpson 2017, p. 197).[2] Following Black feminist scholar Jennifer Nash, generative refusals of the status quo also include movements toward another (Nash 2019). This is because a break with the "what is" might facilitate a movement toward others with an openness to transformative change and to be transformed and changed by another (Nash 2019). As Stonefish guides us, movements toward another move in relation to the universe, which includes land, water, sky, and all human and non-human beings.

Building on the rippling of lived experience as acts of resistance, we turn towards Stonefish's guided movement creation activity, "Touching the Universe with Love" that emerged during the development of our learning platform. We present this example as a powerful source of creation for us because it offers critically important insights into ableist-colonial interference in Anishinaabe political orders and the generative refusal of Nishnaabeg as expressed through their embodied actions and philosophies of difference. Through her guided movements, Stonefish's living experiences emerged as a powerful source of transformative learning within a Nishnaabeg ethic of noninterference that generated mutual respect and a space of collaborative and sustainable world-making.

The following transcript of a movement activity led and guided by Stonefish during our 18 January 2022, team meeting articulates the power and creative potential of moving within a mixed ethical space to activate non-assimilative ways of creating together to interfere in colonial eugenic logics and shift our experiences of and relations to local ecosystems.

Well, before white settlers came here, this continent was one of love, one of understanding. When we extend our hands when we give our thanksgiving, we always touch the earth, and we touch the water. We do not necessarily just extend our hand to the front, but we always extend our hands to the back as well because there might be little children or people who are like myself that cannot walk real fast. You still let them know that we are of some worth, and everybody is of worth here. Since we are here at this time, and we share this planet together.

So, when we bring our hands together to touch the earth, we also go down to the floor, to the earth.

Since movement is so important, especially throughout your whole life, you should be able to be mobile, you should be able to move around. Even for my granddaughter. I

want to tell a story about her because when she was in high school, they have a thing called IEP—an individual education plan—but they had her sitting in a wheelchair facing the wall. So, I would go to the school. I volunteered, so I could help all the children who needed that extra help, to lift their hands, to move all their little fingers and all their bones—just move them around, because we all need that. We all need to feel that safe touching. We need to feel our mother, the earth. We were not meant to just forget about one another, but to come together in a circle as we look at our mother, the earth. So, if you could stand up, and you bring your hands up like this to your own body yourself, and then you go to the floor. You ball yourself up like the earth too, like how she does. You lower your head. Put your head far down on the ground.

Then you touch your toes, massage your toes, because your feet are the ones that carry the whole weight of your body. Our mother the earth carries all of our weight and carries everything, all the unnecessary things that are happening to her right now. She is in a lot of pain.

You come up to your hands. You move every little bone in your hands. You go to your left. You make the motion like in a circle. You go far back as you can behind you, and you make this circle with your hands–every finger spread out. This keeps you in balance.

Now you turn to the other side, and you do it the same way. You reach your hands out in the back. In doing so, we say that we are going to leave this planet the way we found it, or better than we found it, for the sake of the next generations to come. So, when we give our thanksgiving, we give thanksgiving for our ancestors and for those who are not yet born, and we keep our hands rolling in front of us again—almost in a swimming motion.

Then we un-cocoon ourselves, and we reach up into the sky world, the stars, and to the moon, and all those other things that are there that we may not know about. Yet, they are what sustains us here.

Then you put your hands, not together, but you intertwine your fingers like this. You put your fingers out. You bring them into your abdomen. You breathe in. You make some sounds of a rushing lake, always keeping your hands by your abdomen because when we have challenges in our life, that is where it affects us—in our abdomen and then our lower backs.

So now we cross our hands over to our shoulders. We feel the weight of our mother, the earth, because we held a lot of things that we should not be holding. We bring that out to our fingertips, and we shoo it out for the universe to take, to re-energize ourselves again always in a circular motion.

We put our hands over our faces, making sure that they are both on the left and right cortex of our brains and the front and back of our brains as well. We touch our ears so that we can hear the birds singing, so that we can hear the cries of the needy, so that we can hear the joyful laughter of those who are happy. Then we take two fingers and put them in front of our ears, like in our high cheekbone area, then we bring them over into our eyes. Whether we can see or not, feel the eye sockets that we all bestow.

Then you touch your nose, touch your nostrils. Then you bring your fingers down and you touch your mouth. You touch your chin. You say to the strongest muscle in your body, the tongue, that I watch my tongue and what I say, because I may hurt somebody. That is the strongest muscle in our body, and that is the one that is most hurtful.

That is why, there are some to remind us, to give us the patience, the love, and the understanding for those who cannot speak words, who are nonverbal. Then we go to our voice box and say not to waste our voice box with anger, not to say things that are going to hurt other people, because we are all a part of creation.

Now we lift our hands up high. Then we put another circle and touch our bosom, and for those of us who are life givers, like our mother, the earth, we were given breasts so that we could nourish our children. We have to give thanksgiving for that. We touch our abdomen again. Then we go to our navel. Where we have carried life, and if we have not carried life, you are an auntie to many, many children of the world, and always remember

that and see the beauty in yourself and the beauty that Mother Earth, the moon, and we as co-creators of the universe still bestow today.

With that, I'm going to caress my hair again. Since this is what they tried to take away from me. Yet I still have it.

Again, I put my hands out like I have the world in my hands, and we do have the world in our hands, and we must be mindful of that. The decisions that we make today are the decisions that are going to either harm or be helpful for future generations to come.

Okay now we have spoken, and now we can relax our bodies, carry on with our business, and take a break. I thank you for taking this time to touch the earth.

Stonefish 's guided movement activity was embedded with the knowledge of bodies moving towards connection to the universe, and how our actions matter. Stonefish reminded us of how settler colonial structures and processes interfered with Anishinaabe practices of touching the universe with love—an ongoing violence that continues to the present day. In placing and leaving Stonefish's granddaughter (Sky) in a wheelchair facing the wall, settler education enacts erasure by refusing mutual respect and self-determination and by preventing her from doing-knowing-creating relationally and communally. Stonefish contrasts this with the embodied practices that she and Sky mutually engage in including how she lovingly moves Sky's hands, fingers, and bones to help Sky activate her body-world of experience and relationship. Stonefish's loving touch and Sky's loving acceptance and response to/return of that touch, they together enact a doing-knowing-creating that generates life-affirming ethical effects. As a creative, non-assimilative, and life-affirming methodology of resistance to colonial eugenics in education, Stonefish guided the team in creating a safe, inclusive space where we could "combin[e] potential energy and kinetic energy as a creative force" (Simpson 2017, p. 183). Through her touch and her movements, she was accessing the entire universe through her practice. Her lived wisdom guided our movements towards affirming the vitality of our unique embodiments while also opening our embodiments towards new understandings of the ways our non-assimilated bodies are central to direct decolonial and counter-eugenic engagement and change.

Through our entangled movements towards decolonization, accessibility, and justice, the world of our digital co-created space "materialized differently through different practices" (Barad 2006, p. 89). As we moved together toward the universe with love, our bodies also opened to being transformed by each other through solidarity that celebrates difference. Opening towards Stonefish's guided movement activity created felt-sensed interferences in the ways we each experience our bodies differently and drew greater awareness to the lived experience of thriving despite the violence of colonial eugenics. The ripple effect of Stonefish 's lived wisdom and resulting decolonizing interferences reverberates into this writing and continues to move at varying tempos through our bodies—traveling through flesh, holding up at bones, and passing through watery, stretchy connective tissue pathways.

**Conclusions**

In this paper and our larger *ITL* project, we aim to model ethics of listening, feeling, sensing, and moving differently, to learn about obligations of mutual care and respect, and how to collectively hold dissenting views in generative ways for more just and sustainable futures on Nishnaabeg and crip survivor-activists' terms. Our process for interacting creatively is informed by Stonefish's and other survivor-activists' lived knowledge of Canadian colonial eugenic interference as well as resistance to it. Simultaneously, their lived experiences as decolonizing anti-eugenic interferences have had life-affirming and transformative effects on our creation process. They continue to inform our doing-knowing-creating in ways that bring bodies into proximity without folding into sameness. Guided by the lived wisdom of survivor-activists, our movements together have developed into a particularly potent and meaningful methodology for interfering with colonial eugenic ideas and practices while also maintaining an ethic of non-assimilative mutual respect for difference. Throughout ITL, lived fleshy interferences of survivor-activists have reverber-

ated between us. Moving, feeling, sensing, and opening toward one interference, then another and another. This is our choreography of co-resistance and our rhythm-relation into the writing of this work as actions of mattering and world-building centered on and across differences. As Stonefish has reminded us, "we do have the world in our hands, and we must be mindful of that. The decisions that we make today are the decisions that are going to either harm or be helpful for the future generations to come".

**Author Contributions:** Conceptualization, E.K., C.R. and M.S.; Methodology, E.K., C.R. and M.S.; Formal analysis, E.K., C.R. and M.S.; Investigation, E.K., C.R. and M.S.; Writing—original draft, E.K.; Writing—review & editing, E.K., C.R. and M.S.; Supervision, C.R.; Project administration, E.K.; Funding acquisition, E.K. and C.R. All authors have read and agreed to the published version of the manuscript.

**Funding:** eCampusOntario: GUEL—564; University of Guelph: Learning Enhancement Fund; Social Sciences and Humanities Research Council's Partnership fund #895-2016-1024.

**Institutional Review Board Statement:** This article describes a co-created knowledge creation and dissemination project series. The participants, including the authors of this article, collectively created methods of working together and the projects' outcomes.

**Data Availability Statement:** Not applicable.

**Conflicts of Interest:** The authors declare no conflict of interest.

## Notes

1. Tsimshian Art History scholar Mique'l Dangeli uses the term "dancing sovereignty" to describe song and dance performances that activate protocols "in ways that affirm hereditary privileges [ . . . ] and territorial rights to land and waterways" (Dangeli 2016, p. 75).

2. Examples include forcing First Nations onto reservations and then creating a pass system (between 1885 and 1945) that made it illegal to move outside of ones reserve without a pass (Cram 2016), forced relocation of First Nations children to residential schools, and banning traditional and ceremonial dancing. The so-called potlatch ban, for example, could result in six months of jail time for those found guilty of participating in dances and festivals (Anderson 2018). Yet, First Nations continue to remember and embody power in their language, culture, and tradition, including their dances and movements (Anderson 2018).

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
