# Peer review of "Towards Decolonial Choreographies of Co-Resistance"

_socsci, doi:10.3390/socsci12040204_

Round 1
Reviewer 1 Report
This article details an ambitious and far-reaching project that makes an important contribution to decolonializing methods and studies in the arts and humanities, with particular implications for somatics, dance and disability studies. The degree of integrated, discursive praxis between Indigenous and settler artists, scholars and artist-scholars is unprecedented as are the multi-layered approach to knowledge mobilization amongst Indigenous settler communities.
Unfortunately the writing in the first seven pages, is extremely dense and does not foster a cohesive progression or tacit acknowledgement of the unique layering of theory from multiple disciplines. Theoretical considerations and interventions frequently circle back on themselves and rely on universalizing statements that occur at the beginning of a paragraph. I strongly recommend that, following an introduction to the ITL project including how and why the project was initiated, what it hoped to achieve, as well as its the theoretical underpinnings and many outputs, that the article begin at Section 4. Sections 1-3 could be moved to the end of the article as part of an integrated, discussion of the significance, impact and future implications of this very important work. Sections 1-3, in their current place and densely written form, prevent the reader from assessing the real, tangible value of this project for themselves. Please foreground the project, its contributors and objectives, then explain to the readers how the project resists, inverts, confronts, so many Eurocentric, ableist, inherently racist, intellectual notions and policies.
In addition, please consider adding/expanding on the following
- please provide context for where the research took place for a readership unfamiliar with Ontario, Southern Ontario, etc., and the structure of its public education and training systems.
- please explain when/why authors choose to invoke Indigenous, Original Peoples, First Nations, in the text.
- please explain and reference "creative coalition" and "choreographies of resistance." The latter is referenced later in the text but needs to appear the first time it is mentioned.
- please explain "residential institution." Does this include Residential Schools, training schools, poor houses and asylums?
- the authors could also consider explaining their approach curatorial practice and their investment in exhibition as research.
Author Response
Response to Reviewer Report One
“Towards Decolonial Choreographies of Co-Resistance”
Reviewer 1
Reviewer 1 recommends that after our introduction, we move straight to section 4 (where our examples begin).
We have made this shift and believe it reduces the initial density of the writing and allows for more progression and layering of theory. Ideas become folded in when they are needed as opposed to all at the start. This also reduces the amount of circling back on theoretical concepts.
Instead of moving Sections 1-3 to the end of the article, we have woven these sections into the examples in alignment with other reviewer suggestions and comments.
In addition, we added/expanded on the following reviewer suggestions (in bold):
- please provide context for where the research took place for a readership unfamiliar with Ontario, Southern Ontario, etc., and the structure of its public education and training systems.
We have added the following text near the bottom of page one:
“Southern Ontario is an important site for this work as it is often wrongfully assumed by many settler Ontarians that race betterment has only occurred in distant times and places. Instead, as central to the Canadian nation-state-in-formation’s land occupation, southern Ontario has been an epicenter of white supremacist race betterment through strategic state interventions in families and the forced placement of individuals from groups targeted for elimination in socially engineered residential environments including Indian Residential Schools, training schools, and reformatories.”
- please explain when/why authors choose to invoke Indigenous, Original Peoples, First Nations, in the text.
In paragraph one, Kelly added a brief qualifier to Original Peoples in consultation with Stonefish. The qualifier focuses on Original Peoples (First Nations) of this hemisphere. We do not use the term “Indigenous.”
- please explain and reference "creative coalition" and "choreographies of resistance." The latter is referenced later in the text but needs to appear the first time it is mentioned.
In paragraph one, we moved up some of the explanation and reference to “choreographing co-resistance” that appeared in section 3.
We’ve added the following language to clarify the phrase creative coalition when it is first used in paragraph one – “works in solidarity across our differences.” We also added an important reference for our work together— the 2012 article by Rubén A. Gaztambide-Fernández called “Decolonization and the Pedagogy of Solidarity.” And we added the following sentence to a fuller explanation that appears in the second paragraph on page 6, which gets at the feeling of our working together. “We access and express that solidarity through the feelings and experiences that emerge from our creating together (Gaztambide-Fernández 2012).”
- please explain "residential institution." Does this include Residential Schools, training schools, poor houses and asylums?
This particular use of “residential institution” describes the many types of institutions designed to contain groups targeted by agents of “race betterment.” It does include Indian Residential Schools, training schools, poor houses and asylums and this has now been clarified in the first paragraph of Example 1.
- the authors could also consider explaining their approach to curatorial practice and their investment in exhibition as research.
In paragraph two, we have added some details regarding the co-curatorial decision to prioritize lived experiences and why we made that choice. We also added details about survivor-activists making specific choices regarding what they share publicly.
Kelly, Stonefish, and Rice have contributed to the revisions, corrected minor typos and have read the revised draft uploaded here.
Reviewer 2 Report
I like the article my only issue is that it was a bit confusing to read--so re-structuring would help. Too much information in the introduction and not a clear description of the way in which the paper would evolve. The re-structuring is minor.
The content is very good. The comments I uploaded should be shown to the author.

Author Response
Response to Reviewer Report Two
(“Towards Decolonial Choreographies of Co-Resistance”)
Reviewer 2
Reviewer 2 recommends re-structuring to add clarity and reduce the amount of information in the introduction.
We have made this shift by weaving sections 1-3 into the examples. Ideas become folded in when they are needed as opposed to all at the start.
Kelly, Stonefish, and Rice have contributed to the revisions, corrected minor typos and have read the revised draft uploaded here.
Reviewer 3 Report
I enjoyed reading this article immensely. It is written in a fluid and engaging manner that guides the reader through the research journey. I recommend it for publication, without hesitation. I have a few suggestions for revision, which do not concern content or contextualising the research within previous scholarship, but aim to ensure that the authors' contributions and implications are foregrounded for the reader.
1. As I was reading, I was excited to learn about what the Into the Light work looked, sounded, and felt like. My experience of reading the article was that I had to wait quite a while, as the lovely examples do not come until p. 7 of the article. I thought that was too late, and would suggest that the authors might consider how to at least introduce these examples earlier in the article, if not move some of them forward.
2. Related to the above comment, the authors seemed to get somewhat bogged down in explaining how eugenics is "difference-destroying," the potential of movement, and the methodologies employed in the research. Again I wondered if there was a way to integrate some of this discussion into the analysis of the examples OR if the authors would consider condensing this explanatory work. It's necessary, but perhaps doesn't need to be so detailed.
3. This is a question that probably springs from my ignorance, but I wanted to know a bit more about how "choreographing co-resistance" fits in to performance and dance studies, as the examples in the article seemed to be less about performance and more about how a thoughtful (and decolonizing?) approach to movement was integrated into the exhibition, team process, and the movement of sustained Anishinaabe resistance across past, present, and future. I would be very interested to know more about this.
Author Response
Response to Reviewer Report Three
(“Towards Decolonial Choreographies of Co-Resistance”)
Reviewer 3
Reviewer 3 recommends re-structuring and moving the examples earlier in the article. Reviewer 3 also recommends integrating some of the discussion from sections 1-3 into the analysis of the examples.
We have addressed both suggestions by weaving sections 1-3 into the examples. Ideas become folded in when they are needed as opposed to all at the start. In the process of integrating the discussion into the analysis, we were also able to reduce some repetition.
Reviewer 3 also wanted to know more about how "choreographing co-resistance" fits in to performance and dance studies “as the examples in the article seemed to be less about performance and more about how a thoughtful (and decolonizing?) approach to movement was integrated into the exhibition, team process, and the movement of sustained Anishinaabe resistance across past, present, and future.”
We appreciate the reviewer’s suggestion to discuss our project more explicitly in relation to dance and performance studies. We have added some text to help readers understand this article in relation to the broader interdisciplinary fields of dance and performance studies with the following text at the end of the introduction: “dance and performance studies one that disrupts a focus on conventional western theatrical notions of dance and performance.” We have also added citations to work that disrupts in this way.
Kelly, Stonefish, and Rice have contributed to the revisions, corrected minor typos, and have read the revised draft uploaded here.